# Calcium Carbonate Nanoparticles Can Activate the Epithelial–Mesenchymal Transition in an Experimental Gastric Cancer Model

**DOI:** 10.3390/biomedicines7010021

**Published:** 2019-03-19

**Authors:** Marina Senchukova, Olesya Tomchuk, Elena Shurygina, Sergey Letuta, Eskender Alidzhanov, Hike Nikiyan, Dmitry Razdobreev

**Affiliations:** 1Department of Oncology, Orenburg State Medical University, Orenburg 460000, Russia; 2Department of Histology, Cytology, Embryology, Orenburg State Medical University, Orenburg 460000, Russia; tom-chukk@ya.ru; 3Department of Pathology, Orenburg State Medical University, Orenburg 460000, Russia; shuryginalena@mail.ru; 4Department of Biophysics and Condensed Matter Physics, Orenburg State University, Orenburg 460018, Russia; letuta@com.osu.ru (S.L.); ekaalid@yandex.ru (E.A.); nhike@yandex.ru (H.N.); leks@unpk.osu.ru (D.R.); 5Institute of micro- and nanotechnology, Orenburg State University, Orenburg 460018, Russia

**Keywords:** calcium carbonate, carcinogenesis, epithelial–mesenchymal transition, gastric cancer, nanoparticles

## Abstract

Previously, we have shown the possibility of intramucosal gastric carcinoma induction by the intragastric administration of a mixture of formaldehyde and hydrogen peroxide in rats. In this study, we report a sizable increase in carcinogenic properties of the mixture when a suspension containing calcium carbonate nanoparticles was added to it. This technique allowed us to reduce both the number of the carcinogen administrations from twelve to two and the time to the cancer induction from six to four months. Although the induced tumors were represented by the intramucosal carcinomas, they were characterized by the extensive invasion of individual tumor cells and their clusters into the muscle layer and serosa as well as into the omentum and blood vessels. Considering that the invasive tumor cells were positive for vimentin, Snail and TGF-β2, we concluded that their invasion was the result of the activation of epithelial–mesenchymal transition (EMT) mechanisms. Thus, taking into account the data obtained, it can be assumed that under the conditions of inflammation or carcinogenesis, the calcium carbonate nanoparticles may affect the activation of EMT mechanisms.

## 1. Introduction

Gastric cancer (GC) is the fifth most common cancer in the world and has the third highest mortality rate [1,2]. Most cases of GC are diagnosed at an advanced stage with metastasis, and thus have poor prognosis [2,3]. In this regard, studying the mechanisms of GC progression is an urgent task and experimental modeling plays an important role.

According to modern concepts, epithelial–mesenchymal transition (EMT) is a crucial process for malignant tumors metastasis which is still the major cause of cancer-related deaths [4,5]. The inducers of EMT may be act hypoxia, oxidative stress, inflammation and other factors [6]. Recently, it has been shown that nanoparticles (NPs) can also stimulate metastasis [7], by increasing the level of TGF-β, one of the key EMT inducers [8]. Within this context, the role of NPs in tumor progression is being studied with increased intensity.

Calcium carbonate (CaCO_3_) is the very abundant mineral in nature. Calcium carbonate-based nanomaterials are widely used in the food and pharmaceutical industries. In a number of studies the cytotoxic effect of CaCO_3_ NPs was noted [9,10], but the probability of its influence to tumor progression was not examined.

Previously, we have shown the potential of GC induction by intragastric administration of a mixture of formaldehyde (FA) and hydrogen peroxide (HP) in white outbred rats. The animals were given a 1.0 mL mixture of 0.4% FA and 0.4% HP (1:1) by oral gavage on the first and third day every two weeks for six months [11]. Further development of this technology led to the creation of a new experimental model of GC with some interesting features associated with EMT.

## 2. Methods and Materials

### 2.1. Animals and Housing

Wistar male rats, 1.5–2.0 months in age and weighing 200–250 g, were supplied by Rappolovo (Leningrad region, Russia). Five animals per cage were housed in the animal facility of Orenburg State Medical University in an air-conditioned room with a 12 h light/dark cycle at 21–22 °C, and were provided ad libitum with a standard commercial rodent food and water. 

### 2.2. Ethics Statement

All experiments and procedures were approved by the Institutional Animal Care Committee of Orenburg State Medical University (application No. 105, approved on 1 October 2014). All animal experiments were performed in accordance with the Guidelines for the Care and Use of Laboratory Animals for Scientific Purposes. All efforts were made to minimize discomfort in these noninvasive procedures.

### 2.3. Experimental Design

Animals were divided into 2 groups, experimental (*n* = 15) and control (*n* = 3). In order to induce GC, the rats were given 1.0 mL of a suspension of CaCO_3_ at a concentration 0.05% in a 1:1 (*v*/*v*) mixture with 0.4% FA and 0.4% HP, by oral gavage on the 1st and 3rd day of the experiment. The suspension was prepared by adding 10 g of CaCO_3_ to 40% FA to a final volume of 100 mL. One mL of this suspension was diluted with 99 mL distilled water and this was then mixed in a 1:1 (*v*/*v*) ratio with a 0.4% solution of HP (2 mL of 3% HP + 13 mL of the distilled water). The resulting solution was administered to animals for 15–20 min. The control animals were given 1.0 mL distilled water in a similar manner and at the same time. Five experimental rats and one control rat were sacrificed at 16 weeks, and the remaining 9 experimental and 2 control rats were sacrificed at 24 weeks. Rats were euthanized using ether vapor. A thorough necropsy of each rat was made, and the entire gastrointestinal tract was scrutinized for lesions. The stomach was opened along the greater curvature and carefully examined. On laparotomy, the organs were also examined for possible metastasis. The number, size and location of lesions were documented. 

### 2.4. Histological and Histochemical Assay

The specimens of gastric lesions with the adjacent mucosal tissues were fixed in 10% neutral buffered formalin for up to 24 h and embedded in paraffin. Sections of 3–5 µm thickness were then cut from the paraffin blocks and stained with Mayer’s hematoxylin and eosin (H&E). 

To image calcium deposits in sections, the von Kossa (calcium stain) technique was used (Abcam, Cambridge, MA, USA), according to the manufacturer’s protocol. Sections were deparaffinized in toluene and hydrated by serial washing in graded ethanol and distilled water. Next, the sections were exposed to 3% silver nitrate under ultraviolet light for 30 min, followed by careful washing with distilled water 2 times. Thereafter, the sections were treated with 5% sodium thio-sulfate for 2–3 min, washed thoroughly with distilled water, counterstained with nuclear fast red solution for 5 min and dehydrated in the usual manner. Histological slides were studied by light microscopy, with an Optika B-350 microscope, connected to a ScopeTek DCM500 camera (Optika Srl, Ponteranica, Italy). 

### 2.5. Immunohistochemistry

For immunohistochemistry (IHC), 4-µm sections were dewaxed, hydrated and heat-treated in 1 mM EDTA pH 8 in an antigen retrieval PT module (Thermo Fisher Scientific Inc., Waltham, MA, USA) at 95 °C for 20 min. Sections were then stained with the following antibodies: Rabbit monoclonal antibody to vimentin (Clone SP20) 1:200 dilution (Spring Bioscience Corp., Pleasanton, CA, USA); goat polyclonal antibody to Snail (ab53519), 1:200 dilution (Abcam, Cambridge, MA, USA); mouse monoclonal antibody against TGF-β2 (220ct16.4.3.1), 1:40 dilution (Cell Marque Corp., Rocklin, CA, USA). The staining procedure was performed according to the manufacturers’ protocols using an Autostainer 480 (Thermo Fisher Scientific Ltd., Vantaa, Finland). The visualization system included UltraVision LP Detection System HRP Polymer & DAB Plus Chromogen (Thermo Fisher Scientific Inc., Waltham, MA, USA) and hematoxylin counterstaining. For the negative control sections, primary antibodies were replaced with phosphate-buffered saline and processed in the same manner. Stained slides were evaluated by two pathologists (MS and OT). The number of Snail and TGF-β2 positive cells was counted in 10 randomly selected fields by light microscopy at 400× magnification. The percentage of positive cells was expressed relative to the total number of tumor or epithelial cells counted.

### 2.6. Detection and Characterization of Calcium Carbonate Nanoparticles 

The sizes and the proportion of nanoparticles (NPs) in the working solution were determined by dynamic light scattering (DLS) on the Photocor Compact-Z analyzer (Photokor Ltd., Moscow, Russia). Before the measurements, the working solution was filtered using a Vladipor membrane filter (MFAS-OC-2 series; ZAO STC “Vladipor”, Moscow, Russia) with a pore size of 450 nm and was placed in a quartz cuvette measuring 1 × 1 × 3 cm for testing. The measurements were carried out at room temperature in manual mode at a scattering angle of 90°. A total of 56 measurements of working solution samples were made.

The morphologic and particle size examination of CaCO_3_ NPs was conducted by SMM-2000 atomic force microscopy (JSC “Proton-MIET Plant”, Moscow, Russia) operated in contact mode. V-shaped silicon nitride cantilevers (MSCT; Bruker Corp., Billerica, MA, USA) with a spring constant of 0.01 N/m and 10 nm tip radius were used. 

Ten μL of the suspension of CaCO_3_ in a mixture of FA and HP was applied to a fresh shear of mica. It was left for a day until the solvents had completely evaporated, after which scanning was performed.

### 2.7. Statistics

STATSTICA 6.0 software (StatSoft Inc, Tulsa, OK, USA) was employed and all data are presented as means ± standard deviation (SD) or medians. Student’s *t*-test was performed for comparisons between two groups. A value of *p* < 0.05 was considered statistically significant.

## 3. Results

The mortality rate among treated rats was 6.7%. Four months from the beginning of treatment, 4 out of 5 rats in the experimental group had gastric tumors, while after 6 months all 9 experimental rats showed malignant tumors of the stomach. The tumors were represented by intramucosal carcinomas and observed only in the stomach body. Macroscopically, the gastric lesions looked like a diffuse thickening of the organ wall. In most cases, multifocal damage of the organ was found. Nodal indurations of gray color were also often observed on the gastric visceral peritoneum and in the omentum. At these times, formation of tumors was not noted in comparable controls.

### 3.1. Pathology of the Gastric Tumor

In accordance with the Vienna classification, all changes detected in the stomach corresponded to gastrointestinal epithelial neoplasia [12]. However, the lesions in the pre-stomach and in the glandular stomach had a different character, as follows. 

In the forestomach, there was an increase of thickness and the number of epidermal rows, mainly at the expense of the basal and spinous layers. Here, polymorphic chaotically located cells with many mitotic figures were observed in large numbers. These changes were most evident in the area of the groove between the rat forestomach and glandular stomach (Figure 1a,b) and corresponded to low grade gastric dysplasia (Category 3 in the Vienna Classification of epithelial neoplasia [12]).

In the stomach body, the lesions corresponding to intramucosal carcinomas (Category 5 in the Vienna Classification of epithelial neoplasia [12]) were observed. The changes mainly affected the basal parts of gastric glands, where areas with marked cytological and architectural atypia were observed. In these areas, the cystic-distorted glands (Figure 2a) often with proliferations of atypical epithelium in their lumen (Figure 2b) were found. These glandular-like structures were lined by flattened polymorphic cells with hyperchromic nuclei of irregular shape and with а large number of bizarre mitotic figures, as well as a markedly increased nuclear/cytoplasmic ratio. These cells were positive for Snail and TGF-β2 (Figure 2c,d). In control animals the single Snail positive cells in the muscle plate and the weak TGF-β2 expression in the chief cells of the gastric glands were observed (Figure 2e,f), which corresponds to results obtained by other authors [13].

The density of Snail positive cells and TGF-β2 positive cells in the lining of the cystic-distorted glands and in the normal gastric glands was 12.5 ± 8.7% and 0% (*p* < 0.01), and 100% and 24.7 ± 16.4% (*p* < 0.001), respectively. In adjacent tissue, the invasion of curved basophilic fibroblast-like cells in the muscle plate and gastric submucosa was noted. These cells were positive for vimentin and Snail. 

In the pyloric glands, the dysplastic changes were more marked in the generative zone of gastric glands, where the ones were lined by prismatic cells with large polymorphic nuclei, having irregular hyperchromatism and irregular nucleoli as well as increased mitotic figures (Figure 3a). Dysplastic changes in the pyloric glands were characterized by the appearance of light vacuoles in the cells. In some areas, the glands were lined by cells with large cytoplasmic mucus vacuoles and hyperchromatic and depolarized nuclei (somewhat resembling signet ring cells), and these were positive for staining with Snail antibodies (Figure 3b). The described changes corresponded to high grade dysplasia (Category 4 in the Vienna Classification of epithelial neoplasia [12]).

The main feature that enabled the classification of relevant changes as malignant tumors, corresponding to intramucosal gastric carcinoma, was the presence of tumor emboli in vessels (Figure 4a) as well as cords and clusters of tumor cells in the muscle layer (Figure 4b), the serosa (Figure 4c), and the omentum (Figure 4d). In these structures, cells of three types were observed: Fibroblast-like cells with a curved shape; cells of oval or polygonal shape with finely dispersed oxyphilic inclusions in the cytoplasm and a small, rounded, centrally located nucleus (Figure 4e); and cells with large, polymorphic, hyperchromic nuclei and with a markedly increased nuclear/cytoplasmic ratio (Figure 4f). 

Immunohistochemistry demonstrated that vimentin (Figure 5a,b), Snail (Figure 5c,d) and TGF-β2 (Figure 5e,f) positive cells were detected in the described structures.

Six months from the beginning of the experiment, two rats had cecum cancer consistent with well-differentiated adenocarcinoma (Figure 6a,b). In four animals, dense lymph nodes were also detected in the mesentery, in the area of the ileocecal angle. Lymph nodes were increased to 0.5–1.2 cm in diameter, but we did not reveal obvious histological signs of their metastatic lesion.

When samples were stained by the von Kossa technique, multiple microcalcifications were revealed in the gastric mucosa of animals from the experimental group, even six months after the beginning of the experiment. The largest number of microcalcifications was observed in stomach body and antrum. Microcalcifications had various shapes and sizes (Figure 7a). Separate crystals were also found in the gastric submucosa. Microcalcifications were not found in the gastric mucosa of the control group of rats (Figure 7b). 

### 3.2. Dynamic Light Scattering (DLS) Analysis of the Carcinogenic Suspension for the Presence and Morphology of Calcium Carbonate Nanoparticles

Analysis showed that CaCO_3_ NPs were presented in the carcinogenic suspension. An example histogram, showing the distribution of CaCO_3_ NPs according to their hydrodynamic radii obtained by the DLS method, is shown in Figure 7c.

The NPs with radius of 0.12–37.9 nm (average radius 7.8 ± 10.8 nm; median 0.42 nm) were found in 0.2%–12.2% (3.1% ± 2.6%; median 2.3%) of all particles dispersed in solution, and with radius of 47.0–267.2 nm (average radius 155.3 ± 86.5; median 66.5) were found in 0.14%–75.5% (16.8% ± 15.4%; median 12.4%) of all particles dispersed in solution.

### 3.3. Atomic Force Microscopic (AFM) Analysis of the Carcinogenic Suspension for the Presence and Morphology of Calcium Carbonate Nanoparticles

Images obtained by AFM showed the presence of CaCO_3_ NPs with spherical or oval shapes (Figure 7d). Both individual NPs and their aggregates were observed in the carcinogenic suspension.

## 4. Discussion

The study of the mechanisms associated with metastasis of malignant tumors is of fundamental importance for predicting the risk of metastasis, its prevention and its successful treatment. The solution to this problem is impossible without the development of effective experimental models that best reflect these mechanisms. 

We have previously reported on the feasibility of GC induction by using a mixture of FA and HP [11]. For tumor induction, 1.0 mL a mixture of 0.4% FA and 0.4% HP (1:1 *v*/*v*) were administered to rats using intragastric gavage on the first and third day every two weeks for six months. Six months after the onset of treatment, tumors were observed in 80% of the animals, and after 7.5 months in all survived rats. Initiated tumors were corresponded to intramucosal carcinoma and characterized by invasion of individual tumor cells and their clusters in muscle and serous layer and in omentum, and by the presence of tumor emboli in vessels. These data conformed with results of some clinical studies, showing that gastric high-grade dysplasia can be associated with submucosal and lymphovascular invasion [14,15]. 

The idea of using a mixture of FA and HP to induce GC was based on published data indicating the mutagenic and carcinogenic properties of these substances. In particular, the administration of FA to drinking water for two years led to the appearance of malignant lymphomas and tumors of the gastrointestinal tract in experimental animals [16]. The carcinogenic effect of FA is likely associated with its genotoxic properties and the ability to form DNA–protein cross-links [17,18]. Genotoxic properties and the ability to stimulate neoangiogenesis have been also noted for HP [19,20,21]. When HP and FA are used together their mutagenic properties have been shown to increase [22]. This increase may be due to a number of interesting chemical properties. Serebrennikov and Golovkin (2005) reported that the starting products of FA and HP reaction as well as their complexes were stable in the mixture for a long time [23]. Regardless of the mixed substances ratio (1:1, 1:2 or 2:1), the intermediate and relatively resistant product of their reaction is the peroxide molecule H_2_C(OH)OOH. Later this molecule decomposes to formic acid, water and carbon dioxide.

In the current study, we presented a new experimental model for GC that was developed as a result of controlled changes in the methodology described above. The proposed model was conformed the intramucosal gastric carcinoma with invasion of tumor cells in the gastric submucosa, the muscular layer, serosa and omentum, as well as tumor emboli in the vessels. We believe that the results of the experiment are consistent with the Klein hypothesis that suggested tumor growth and metastasis could proceed in parallel, independently of each other [24]. In addition, the received results lent legitimacy to the point of view of Japanese pathologists, who consider the severe dysplasia based on nuclear and structural atypia was cancer regardless of the presence of invasion [15,25].

Considering that the invasive tumor cells were positive for vimentin, Snail and TGF-β2, we concluded that their invasion is the result of activation of EMT mechanisms. EMT is a transdifferentiation process in which epithelial cells lose their cell–cell contacts and polarity and acquire mesenchymal properties, coupled to the ability to migrate and to invade the surrounding tissues [26,27]. The activation of EMT occurs under the influence of transforming growth factor beta (TGF-β) that is crucial to the regulation of cell proliferation, differentiation, invasion, migration and cancer metastasis [28]. The induction of EMT entails a reduction or loss of the ability to synthesize cytokeratins and E-cadherin by the epithelial and tumor cells, but the appearance of the ability to synthesize vimentin, Snail, smooth-muscle actin and other markers of EMT as well as an increasing expression of the genes encoding matrix metalloproteinases. As a result of EMT, cell–cell contacts are ruptured, cell morphology changes to fibroblast-like and the cells become mobile [27,29,30]. Additionally, in the process of EMT tumor cells acquire the properties of stem cancer cells, which are associated with high aggressiveness and drug resistance of malignant tumors [31,32].

We think that in interpreting the current experiment, the data of Ke et al. (2008) [33] should also be taken into account. They demonstrated that EMT processes are similar, but not identical, to malignant transformation. As such, they argued that a more strict confirmation is required that cell invasion is the result of not only EMT, but also their malignant transformation. In cases of gastric mucosa intraepithelial lesions, the differential diagnosis between a high degree of dysplasia and cancer presents known difficulties [14,34].

One of the principle results of our study was a sizable increase in the carcinogenic properties of the mixture of FA and HP when suspension containing CaCO_3_ NPs was added to it. The change in the experiment methodology allowed us to reduce the number of administrations of the carcinogen from 12 to two and in a shorter period of time to obtain a new experimental model of intramucosal gastric carcinoma, which was associated with activation of EMT mechanisms. 

It should be noted that studying the role of NPs in the development of various diseases is receiving marked attention. It is known that NPs can not only cause chronic inflammatory and autoimmune diseases, but also malignant neoplasms [35,36,37]. Moreover, NPs can influence the mechanisms of tumor progression by stimulating the growth of metastases [7,38,39]. NPs in the size range 50–270 nm are the most dangerous in terms of producing pathological conditions [40]. These sizes allow NPs to easily penetrate between the epithelial cells into the underlying tissues, into the cells and vessels, as well as into the extracellular matrix. Macrophages, lymphoid and dendritic cells are capable of spreading the NPs beyond the sites of primary deposition. Inside cells, the NPs damage mitochondria and can penetrate into the nucleus of cells, causing DNA breakage [41]. In our experiment, when staining the stomach samples by the von Kossa method, we also observed microcalcifications in the cells and the intercellular substance of the gastric mucosa, even six months after the carcinogenic suspension administration (see Figure 7a,b). 

The toxicity of NPs is explained by their physicochemical properties, the catalytic activity of the NP surface and their ability to penetrate through the lungs, skin, gastrointestinal tract into any human organ and tissue, including the central nervous system [42]. The main physicochemical properties of NPs determining their toxicity are absorption, distribution, toxicokinetics, metabolism, and features of their biodegradation. The biological activity of NPs also depends on their size, shape, and surface properties, such as their charge and catalytic properties. A number of authors have shown that the size of NPs is identical to those of signaling molecules and cellular receptors. In addition, nanoparticle sizes correspond to exosome sizes playing an important role in intracellular communication and in delivering molecular signals from one cell to another [43]. The contact of NPs with immune cells results in the following effects:Increased synthesis of pro-inflammatory cytokines NF-κB (nuclear factor kappa B), IL-6 (interleukin 6), TNF-α (tumor necrotic factor-α) and IL-2 [44,45];release of large concentrations of reactive oxygen and nitric oxide with the properties of free radicals that damage cell membranes and genetic material [44,45];development of immunological tolerance due to the increase of TGF-β1serum levels [35,39,44];generation of CD4+CD25+FoxP3+ regulatory T-cells [46,47,48] and the PD-L1 activation [49,50]. Considering that NPs used in the experiments had different chemical composition, it can be assumed that these properties may be associated not only with their structure, but, first of all, their size.

It is important to emphasize that that TGF-β, being one of the key activators of EMT, induces pro-invasive and pro-metastatic immune responses by the cooperation with stem cell pathways, such as Wnt and Ras signaling [51,52]. In addition, TGF-β plays a key role in the development of immunological tolerance. The cytokine inhibits the differentiation of cytotoxic T-lymphocytes, Th1 and Th2 cells, and stimulates the formation of peripheral Treg, Th17, Th9 and Tfh cells in response to various immune stimuli [53]. Increased expression of TGF-β is noted in various inflammatory, autoimmune and oncological diseases [53,54].

Once again, we note that CaCO_3_ is a very abundant mineral in nature. Calcium carbonate-based nanomaterials are widely used in the food and pharmaceutical industries. Some authors noted its very low toxicity during oral administration [55]. However, other researchers have shown its potential cytotoxicity. Kim (2015) demonstrated the calcifying NPs cytotoxicity when they were co-cultivated with different cell cultures [10]. The cytotoxicity was manifested by the NPs endocytosis, the production of intracellular reactive oxygen species, the membrane damage and cell apoptosis [56]. Similar results were obtained by Horie [9].

We believe that it is also necessary to note a number of other interesting features of CaCO_3_ NPs. In particular, CaCO_3_ is the main mineral element of mineral-organic NPs that were described in detail by Martel and Young [57]. These NPs have been established to form in human biological fluids when the concentration of CaCO_3_ and phosphate ions exceeds saturation. Contact of crystalline CaCO_3_ (both endogenous and exogenous) with biological fluids like serum leads to the formation of round amorphous NPs [58,59,60] that had earlier been misconstrued as nanobacteria [61]. Described NPs are found in different human biological fluids, and their presence is associated with various pathological processes, including Alzheimer’s disease, atherosclerosis, cancer, kidney stones, prostatitis and others [58,59,60]. In inflammation the high concentration of mineral-organic NPs in biological fluids can lead to ectopic calcification [58]. 

Powell (2015) described the mechanism of the endogenous formation of calcium containing mineral-organic NPs [62]. They showed that calcium and phosphate ions are secreted from the distal small intestine into the lumen and then react with exogenous and endogenous proteins with the formation of mineral-organic NPs in the intestinal lumen. These proteins transport them to immune cells of intestinal tissue. In wild-type mice, intestinal immune cells containing these naturally formed NPs expressed the immune tolerance-associated molecule PD-L1. The authors suggest that the described mechanism participates in the formation of immunological tolerance against food and the gut microbiota. Given that mineral-organic NPs play a key role in ectopic calcification [58,59,60], it can be assumed that extra-bone calcification, which is observed from atrophic, dystrophic, dysplastic and necrobiotic changes in tissues, as well as in benign and malignant tumors [63,64,65,66,67,68,69], can also be associated with the activation of the immunological tolerance mechanisms. The fact that the same cytokines and mediators are involved both in the processes associated with extracellular calcification and in the development of immunological tolerance [68,69] indirectly supports this assumption

## 5. Conclusions

Taking into account the data obtained and the analysis data of the scientific literature, it can be assumed that under the conditions of inflammation or carcinogenesis, the CaCO_3_ NPs may affect the activation of mechanisms of EMT and immunological tolerance. We believe that further research in this direction is necessary, since knowledge of these mechanisms may be of fundamental importance for the development of new approaches to the treatment and prevention of various diseases, including malignant neoplasms.

## Figures and Tables

**Figure 1 biomedicines-07-00021-f001:**
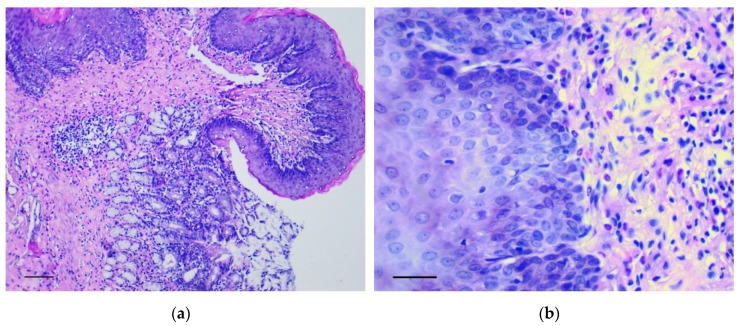
The pathological changes in the forestomach. (**a**) Area of the groove between the forestomach and glandular stomach. Mayer’s hematoxylin and eosin (H&E) staining, scale bar 200 μm; (**b**) Increasing thickness and number of epidermal rows in the basal and spinous layers of forestomach. H&E staining, scale bar 100 μm.

**Figure 2 biomedicines-07-00021-f002:**
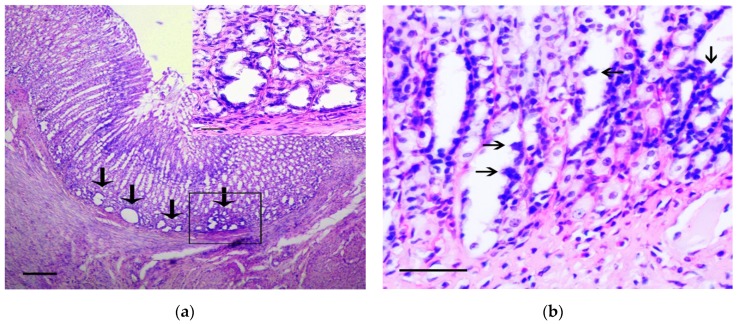
The pathological changes in stomach body. (**a**) Basal part of gastric glands with marked cytological and architectural atypia (arrows). In the box the selected area is shown. H&E staining, scale bar 200 μm (in box—100 μm). (**b**) Proliferations of atypical epithelium in the lumen of glandular-like structures (arrows). H&E staining, scale bar 100 μm. (**c**–**d**) The marked Snail (**c**) and TGF-β2 (**d**) expression in gastric mucosa in rats of the experimental group is shown. Immunoperoxidase staining with antibody to Snail and TGF-β2, scale bar 100 μm. (**e**–**f**) Very weak Snail (arrows) (**e**) and TGF-β2 (**f**) expression in gastric mucosa in rats of the control group is shown. Immunoperoxidase staining with antibody to Snail and TGF-β2, scale bar 100 μm.

**Figure 3 biomedicines-07-00021-f003:**
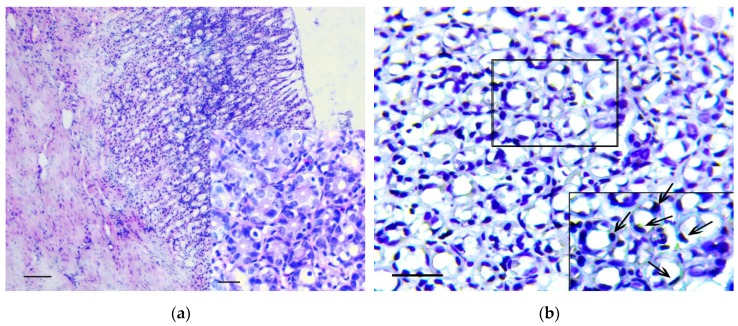
The pathological changes in the pyloric stomach. (**a**) The dysplastic changes in the pyloric glands of stomach. H&E staining, scale bar 100 μm. In the box the polymorphic prismatic cells with large nuclei, having irregular hyperchromatism and irregular nucleoli in generative zone of pyloric glands are shown. H&E staining, scale bar 100 μm. (**b**) Positive reaction to Snail in cells lined the pyloric glands resembling signet ring cells. In the box the selected area with signet ring cells (arrows) is shown. Immunoperoxidase staining with antibody to Snail, scale bar 100 μm.

**Figure 4 biomedicines-07-00021-f004:**
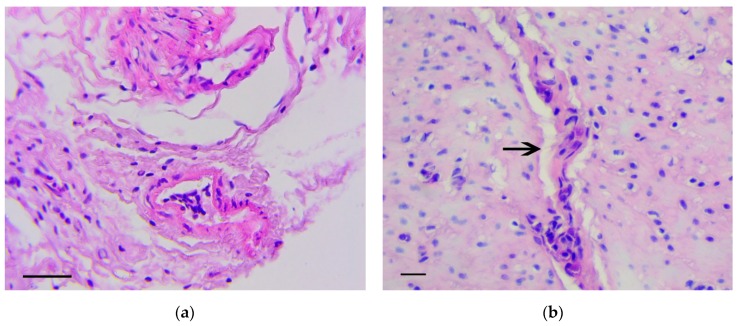
Invasion of tumor cells. H&E staining. (**a**) Tumor embolus in the vessel, scale bar 100 μm. (**b**) Invasion of tumor cells in muscle layer (arrows), scale bar 100 μm. (**c**) Invasion of tumor cells in serosa layer (arrow). In the box the marked area is shown at magnification, scale bar 200 μm. (**d**) Invasion of tumor cells in omentum. In the box the selected area is shown at higher magnification, scale bar 200 μm. (**e**) The clusters of tumor cells in the muscle layer consisting of fibroblast-like cells with a curved shape (1-black arrows) and cells with finely dispersed oxyphilic inclusions in the cytoplasm (2-red arrows), scale bar 100 μm. (**f**) The cells with large, polymorphic, hyperchromic nuclei and with a markedly increased nuclear/cytoplasmic ratio in the muscle layer of stomach, scale bar 100 μm.

**Figure 5 biomedicines-07-00021-f005:**
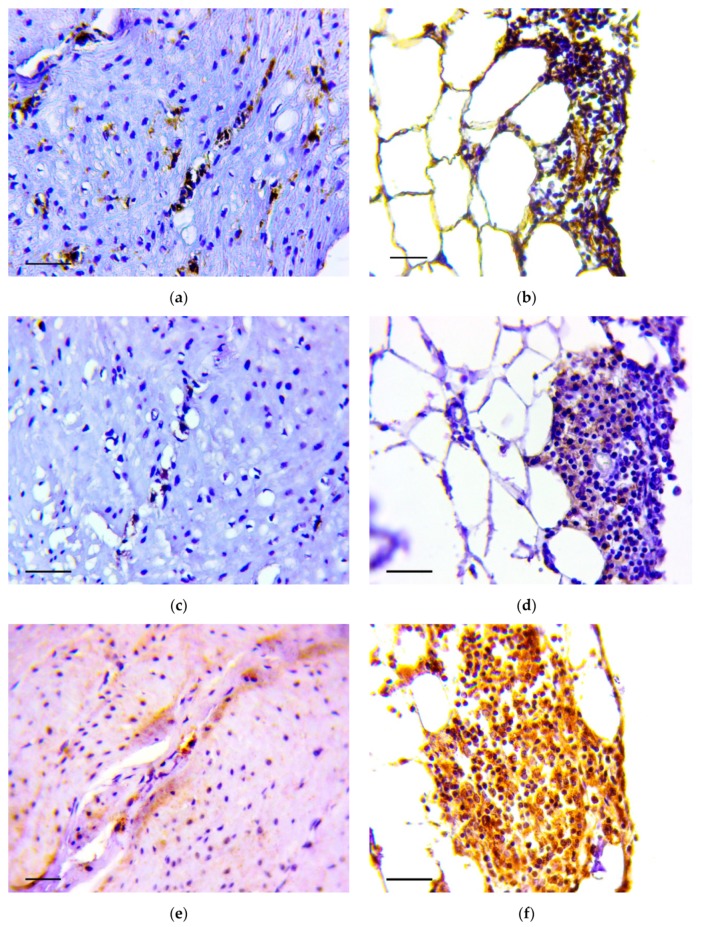
Invasion of tumor cells in the muscle layer (**a**,**c**,d) and in omentum (**b**,**d**,**f**). Positive reaction to vimentin (**a**,**b**), Snail (**c**,**d**) and TGF-β2 (**e**,**f**). Immunoperoxidase staining with antibody to vimentin, Snail and TGF-β2, scale bar 100 μm.

**Figure 6 biomedicines-07-00021-f006:**
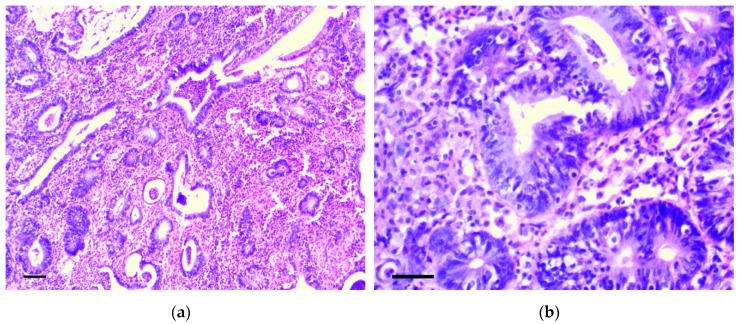
Well-differentiated adenocarcinoma of cecum. H&E staining: (**a**) 200×, scale bar 200 μm, (**b**) 800×, scale bar 100 μm.

**Figure 7 biomedicines-07-00021-f007:**
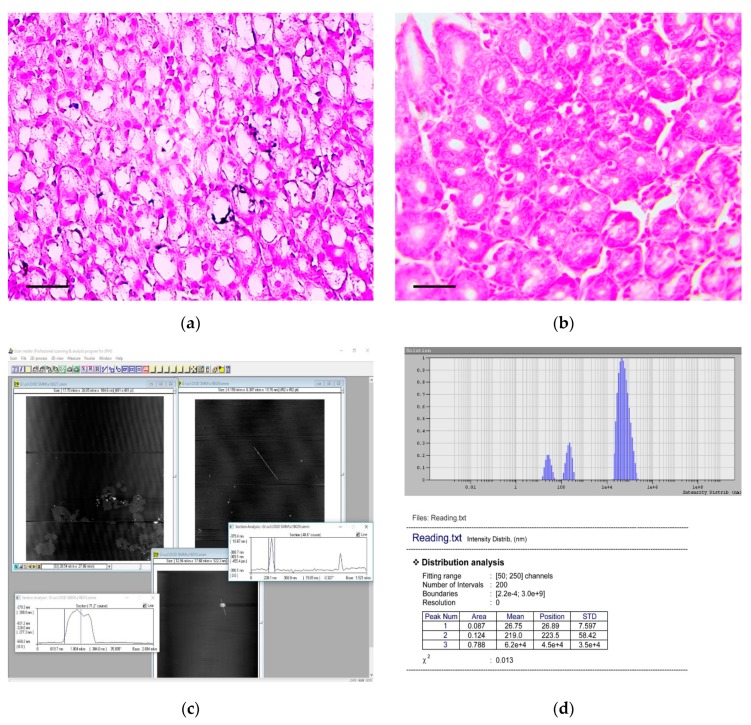
Detection and characterization of calcium carbonate nanoparticles in the carcinogenic suspension and microcalcifications in the gastric mucosa. (**a**) Multiple microcalcifications in the gastric mucosa of experimental rats. von Kossa staining, scale bar 100 μm. (**b**) The absence of microcalcifications in the gastric mucosa of control rats. von Kossa staining, scale bar 100 μm. (**c**) The imaging of aggregates (nanoparticles) obtained by the atomic force microscopy. The scan size is 7.8 × 5.6 μm. (**d**) The histogram of calcium carbonate nanoparticles distribution in carcinogenic suspension according to their hydrodynamic radii, obtained by the dynamic light scattering method.

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
