# Peer review of "Calcium Carbonate Nanoparticles Can Activate the Epithelial–Mesenchymal Transition in an Experimental Gastric Cancer Model"

_biomedicines, 2019, doi:10.3390/biomedicines7010021_

Reviewer 1 Report

Comments for biomedicines-455719,

The aim of manuscript entitled “Calcium Carbonate Nanoparticles Can Activate the Epithelial-Mesenchymal Transition in an Experimental Gastric Cancer Model” by SenchukovaM et al,is to suggestthat under the conditions of inflammation or carcinogenesis, the calcium carbonate nanoparticles may affect the activation of epithelial-mesenchymal transition (EMT) mechanisms in experimental gastric cancer model.

Despite their results might be interesting there are some points which need clarification.

Minor points  
1. Are not indicated histological type, grading and site of tumors (Table?). No correlation with these elements has been made.
2. Authorsshouldspecify in introduction and discussionthe biology and molecular correlation between the conditions of inflammation or carcinogenesis, the CaCO3 NPs, the activation of mechanisms of EMT and immunological tolerance in this study.

Author Response

Dear reviewer, thank you for reviewing our manuscript and equitable comments. In accordance with the comments, we made the following changes to the manuscript:

1. Are not indicated histological type, grading and site of tumors (Table?). No correlation with these elements has been made.

We have included information concerning histological type, grading and site of tumors into the text of the manuscript (line 19, 25, 137-138, 144-145, 154-155, 160-161, 188-189...).

2.Authors should specify in introduction and discussion the biology and molecular correlation between the conditions of inflammation or carcinogenesis, the CaCO3 NPs, the activation of mechanisms of EMT and immunological tolerance in this study

We noted that «The largest number of microcalcifications was observed in stomach body and antrum» (Line 223).

At the same time, we would like to note that the induction of gastric cancer by a mixture of FA and HP without calcium carbonate also initiated intramucosal carcinoma with invasion of tumor cells in muscle and serous tissue and omentum. This suggests that CaCO3 NPs did not initiate gastric cancer, but potentiated the carcinogenic effect of a mixture of formaldehyde and hydrogen peroxide, which was manifested in a reduction in the number of carcinogen administrations and the time to the cancer induction. For this reason, we believe that for the quantitative evaluating the role of CaCO3 NPs in activating the mechanisms of EMT and immunological tolerance it is more objective the comparison of the expression of Snail, TGF-β and other markers in models of gastric cancer induction by the mixture of FA and HP with and without adding the calcium carbonate. We are planning to undertake such study in the near future.

Sincerely, Dr. Marina Senchukova

Reviewer 2 Report

The authors have made an interesting and a complete study of the histology of the gastric tissues demonstrating the treatment of rats with formadehyde, hydrogen peroxide in a mixture containing calcium carbonate is suitable to develop a model of gastric cancer in less time than only using with formadehyde, hydrogen peroxide. The results are interesting and the article is well presented and formatting.

I only have minor concerns regarding:

-The chemical formula of calcium carbonate. The number “3” has to be a subscript. Please correct this throughout the document.

-The same happens in line 39 of the discussion. The empirical formula of the peroxide molecule has a number “2” that has to go in subscript.

- Also it has to be corrected the beta in TFG-2b in figures 2 and 5, it must be a beta with the Greek symbol.

Author Response

Dear reviewer, thank you for reviewing our manuscript and equitable comments. In accordance with the comments, we made the following changes to the manuscript:

We have corrected the chemical formula of calcium carbonate and peroxide molecule

We have corrected the spelling TGF-β2 in figures 2 and 5.

Sincerely, Dr. Marina Senchukova